# Transcriptome Analysis of Air Space-Type Variegation Formation in *Trifolium pratense*

**DOI:** 10.3390/ijms23147794

**Published:** 2022-07-14

**Authors:** Jianhang Zhang, Jiecheng Li, Lu Zou, Hongqing Li

**Affiliations:** School of Life Sciences, East China Normal University, Shanghai 200241, China; 52181300002@stu.ecnu.edu.cn (J.Z.); 51201300006@stu.ecnu.edu.cn (J.L.); 15201799715@163.com (L.Z.)

**Keywords:** *Trifolium pratense*, transcriptome, air space-type variegation, leaf variegation, variegated leaf plants

## Abstract

Air space-type variegation is the most diverse among the species of known variegated leaf plants and is caused by conspicuous intercellular spaces between the epidermal and palisade cells and among the palisade cells at non-green areas. *Trifolium pratense*, a species in Fabaceae with V-shaped air space-type variegation, was selected to explore the application potential of variegated leaf plants and accumulate basic data on the molecular regulatory mechanism and evolutionary history of leaf variegation. We performed comparative transcriptome analysis on young and adult leaflets of variegated and green plants and identified 43 candidate genes related to air space-type variegation formation. Most of the genes were related to cell-wall structure modification (*CESA*, *CSL*, *EXP*, *FLA*, *PG*, *PGIP*, *PLL*, *PME*, *RGP*, *SKS*, and *XTH* family genes), followed by photosynthesis (*LHCB* subfamily, *RBCS*, *GOX*, and *AGT* family genes), redox (*2OG* and *GSH* family genes), and nitrogen metabolism (*NodGS* family genes). Other genes were related to photooxidation, protein interaction, and protease degradation systems. The downregulated expression of light-responsive *LHCB* subfamily genes and the upregulated expression of the genes involved in cell-wall structure modification were important conditions for air space-type variegation formation in *T. pratense*. The upregulated expression of the ubiquitin-protein ligase enzyme (E3)-related genes in the protease degradation systems were conducive to air space-type variegation formation. Because these family genes are necessary for plant growth and development, the mechanism of the leaf variegation formation in *T. pratense* might be a widely existing regulation in air space-type variegation in nature.

## 1. Introduction

Variegated leaf plants are a special group of colored-leaf plants with stable patterns formed by differently colored leaf areas, and are ideal materials for the study of plant chromatology, the mechanism of plants adapting to low-light environments, and the mechanism of chloroplast development. In variegated leaf plants, air space-type variegation is the most diverse, and it is caused by conspicuous intercellular spaces between the epidermal and palisade cells and among the palisade cells at non-green areas [1]. Research on the function of air space-type variegation has a good foundation, and previous reports have found that leaf variegation plays an important role in adapting to low-light environment, resisting low temperatures, and avoiding being foraged by herbivores [2,3,4,5,6].

Studies of variegated leaf plant transcriptomes have mostly focused on the types of variegations caused by chlorophyll deficiency and have analyzed the molecular regulatory mechanism of chlorophyll loss in the variegated areas of plants, such as *Ananas comosus* var. *bracteatus* [7], *Epipremnum aureum* ‘Marble Queen’ [8], *Triticum aestivum* [9], *Clivia miniata* var. *variegata* [10], *Betula platyphylla* × *B. pendula* [11], and *Ficus microcarpa* ‘Milky Stripe Fig Leaf’ [12]. In the transcriptome study of the air space-type variegation in *Arabidopsis thaliana im* mutants and *Paphiopedilum concolor*, the pathways of photosynthesis, redox, and nitrogen metabolism were mainly considered, but the gene expression changes related to cell wall development to maintain plant cell morphology were not discussed [13,14]. Although these studies have provided a deep understanding of the variegation mechanism in plants, the transcriptome of the air space-type variegation is still poorly understood.

*Trifolium pratense* L., belonging to the family Fabaceae, is native to north Africa, southwest Asia, and Europe, and is widely cultivated around the world as an ornamental flowering plant and also for its foraging importance [15,16]. The *T. pratense* genome map was published in 2015, and 40,868 genes and 42,223 transcripts were annotated and anchored to seven chromosomes [17]. Typically, more comprehensive studies were reported for their analysis of the splice isoforms, leaf senescence, and transcriptional regulation in response to silver ions [18,19,20]. Importantly, *T. pratense* leaflets exhibit V-shaped air space-type variegation that has a greater aesthetic appeal and ornamental value [1]. However, apart from the ornamental value of *T. pratense*, the molecular regulatory mechanism of leaf variegation remains unknown. An in-depth study of the molecular mechanism for air space-type variegation formation is not only conducive to revealing its color development principle and summarizing the law of ecological adaptability, but also lays the groundwork for the continued study of the origin and evolution of air space-type variegation.

In the present study, we studied two *T. pratense* phenotypes, “variegated plants” and “green plants”, with distinct leaf coloration features. To thoroughly understand the leaf variegation formation in *T. pratense*, we observed the transverse sections of the adult leaflets and performed transcriptome sequencing analysis of the young leaflets (unexpanded leaflets) and adult leaflets (fully expanded leaflets) of the variegated and green plants. Our findings elucidate the potential regulatory network of air space-type variegation formation in *T. pratense*, which would promote the understanding of the other variegations and their origin and evolution.

## 2. Results

### 2.1. Quality Evaluation and Reliability Verification of RNA-Seq Data

The RNA-seq data of 15 samples of GA1 (the position corresponding to VA1 of green plant adult leaflets), GY (green plant young leaflets), VA0 (green area of variegated plant adult leaflets), VA1 (white area of variegated plant adult leaflets), and VY (variegated plant young leaflets) from five parts of *T. pratense* leaflets were analyzed (Table 1). The raw reads of each sample were between 47.39 and 52.70 million. After removing the low-quality reads, the number of clean reads was between 42.55 and 47.92 million. The RNA-seq of each sample yielded ≥6.38 Gb clean data with 95.85% of bases scoring Q20 and above, and 87.14% of bases scoring Q30 and above. After alignment, 34,771 genes were obtained from *T. pratense*, including 33,226 known genes and 1545 predicted new genes. Pearson’s correlation coefficient between the samples were all >0.8 (Appendix A), suggesting a high reliability of our RNA-seq data. The alignment rates of the samples and *T. pratense* genome were all ≥76.59%, and those of the matched gene sets were all ≥67.25% (Table 1). The alignment rates of each sample were relatively high. Thus, the data met the basic requirements of transcriptome sequencing analysis and could be used for subsequent analysis.

### 2.2. Screening and Identification of Differentially Expressed Genes (DEGs) Related to Air Space Type Variegation Formation

#### 2.2.1. Differential Gene Expression Analysis

Differential gene expression analysis was performed on all of the expressed genes by comparing their expression levels in young leaflets and of the green and variegated plants (GY vs. VY), the position corresponding to VA1 of the green plant adult leaflets and the white area of the variegated plant adult leaflets (GA1 vs. VA1), and the green area of the variegated plant adult leaflets and the white area of the variegated plant adult leaflets (VA0 vs. VA1). In this study, the genes with FPKM ≥ 1, |log_2_ fold change| ≥ 1, and adjusted *p* ≤ 0.001 were defined as significant DEGs to improve the accuracy of the results. As shown in Figure 1a, 5700 DEGs were obtained from the three comparison groups. There were 3654 in GY vs. VY, including 2703 upregulated and 951 downregulated genes. There were 2469 in GA1 vs. VA1, including 1554 upregulated and 915 downregulated genes. There were 1216 in VA0 vs. VA1, including 473 upregulated and 743 downregulated genes. The GY vs. VY and GA1 vs. VA1 comparison groups shared 1107 DEGs, and the GA1 vs. VA1 and VA0 vs. VA1 comparison groups shared 324 DEGs (Figure 1b). These common DEGs received more attention in the analysis of air space-type variegation formation in *T. pratense*.

#### 2.2.2. KEGG Enrichment Analysis

To identify the important metabolic pathways, KEGG enrichment analysis was performed on DEGs that were significantly associated with the three comparison groups. The DEGs of the GY vs. VY comparison group were enriched in 131 pathways, of which five pathways were significantly enriched—flavonoid biosynthesis, isoflavonoid biosynthesis, cutin, suberin, and wax biosynthesis, photosynthesis-antenna proteins, and DNA replication (Figure 2a). These pathways are involved in the cell wall development and photosystem formation, which are necessary for leaf development. The DEGs of the GA1 vs. VA1 comparison group were enriched in 126 pathways, of which five pathways were significantly enriched—plant–pathogen interaction, flavonoid biosynthesis, isoflavonoid biosynthesis, phenylpropanoid biosynthesis, and MAPK signaling pathway-plant (Figure 2b), which are involved in the cell wall development and stress signal transmission in plants; thus, implying that the secondary metabolism of *T. pratense* variegated plant is enhanced in adult leaflets and may suffer the influence of environmental factors. The DEGs of the VA0 vs. VA1 comparison group were enriched in 112 pathways, of which seven pathways were significantly enriched—glutathione metabolism, isoflavonoid biosynthesis, alpha-linolenic acid metabolism, plant hormone signal transduction, flavonoid biosynthesis, carotenoid biosynthesis, and cyanoamino acid metabolism (Figure 2c). These pathways are related to the cell wall development, production, and scavenging of reactive oxygen species, pigment synthesis, the response to external cell stimuli, and the decomposition of substances.

In addition, the DEGs of the three comparison groups were significantly enriched in both flavonoid biosynthesis and isoflavonoid biosynthesis pathways (Figure 2). In plant growth and development, these two pathways are involved in the synthesis of the cell wall and pigments, as well as playing important regulatory roles in the synthesis of lignin, pectin, and cellulose [21,22,23,24,25]. Thus, we focused on screening the key genes of these pathways. The analysis showed that air space-type variegation formation in *T. pratense* was related to photosynthesis, reactive oxygen species level, and cell wall development in variegated plants. Whether the formation was related to changes in the leaflets of variegated plants in response to light or other environmental factors merited further study.

#### 2.2.3. Identification of Genes Related to Air Space-Type Variegation Formation

To further study the regulatory mechanism of air space-type variegation in *T. pratense*, 5700 DEGs involved in the three comparison groups were compared by Mercator software, and the overall comparison rate was 66.44% (Figure 3a), which could satisfy the subsequent MapMan analysis. Through the corresponding Arabidopsis orthologs, a metabolism overview of DEGs in the three comparison groups from *T. pratense* was visually analyzed in MapMan software, focusing on the pathways related to cell wall development, photosynthesis, redox, and nitrogen metabolism [12,13,26]. The results showed that 112, 80, and 62 DEGs were assigned to these four pathways for the GY vs. VY, GA1 vs. VA1, and VA0 vs. VA1 comparison groups, respectively (Appendix A). In the three comparison groups, the DEGs of the pathways related to photosynthesis were mainly downregulated, and the DEGs of the pathways related to cell wall development, redox, and nitrogen metabolism were mainly upregulated (Figure 3b–d; Appendix A). Thus, the air space-type variegation formation in *T. pratense* was related to different expression trends of DEGs in the four pathways. Venn analysis was performed on the DEGs obtained in MapMan analysis; there were 43 common upregulated and downregulated DEGs (Appendix A), which were candidate genes for the air space-type variegation formation in *T. pratense*.

##### Cell Wall Development Related Pathway

Based on the analysis results of MapMan and Venn (Figure 3 and Appendix A), the functions and expression of the 43 common DEGs involved in cell wall development, photosynthesis, redox, and nitrogen metabolism were further studied. The cell wall development pathway holds the majority in 29 common DEGs (Figure 4; Appendix A), involving various mechanisms, such as cell wall protein synthesis, cellulose synthesis, cell wall degradation and modification, and pectinesterase synthesis. There are 11 gene families related to the cell wall structural modification, including reversibly glycosylated polypeptide 2 (*RGP*), FASCICLIN-like arabinogalactan family genes (*FLA*), cellulose synthase-like family genes (*CSL*), cellulose synthase family genes (*CESA*), polygalacturonase-inhibiting family genes (*PGIP*), polygalacturonase family genes (*PG*), pectin lyase-like superfamily genes (*PLL*), expansin family genes (*EXP*), xyloglucan endotransglucosylase/hydrolase family protein (*XTH*), SKU5 similar family genes (*SKS*), and plant invertase/pectin methylesterase inhibitor superfamily genes (*PME*). Among these common DEGs, only the expression of *FLA16* (BGI_novel_G001184) of the FASCICLIN-like arabinogalactan gene family was downregulated, and this gene was annotated as FASCICLIN-like arabinogalactan protein precursor 16, which is related to cell adhesion. The expression of the other genes was upregulated, among which, *CSL*, *EXP*, *FLA*, *PG*, *PLL*, *PME*, and *XTH* are important gene families involved in cell wall development. The upregulated expression of these genes can cause cell wall remodeling and loosen the connection between cells 2733, which was the potential cause of the intercellular spaces’ formation in the variegated area.

##### Photosynthesis Pathway

There were nine common DEGs in the photosynthesis pathway (Figure 4; Appendix A), mainly involving four gene families. The DEGs related to photosynthesis (including light reaction and the Calvin cycle) that encode light harvesting complex B subfamily genes (*LHCB*) and ribulose bisphosphate carboxylase (small chain) family genes (*RBCS*) were all downregulated in both the GY vs. VY and GA1 vs. VA1 comparison groups, indicating that these DEGs were inhibited in both the young and adult leaflets of the variegated *T. pratense* plants, which was consistent with the results of previous studies of leaf variegation in Arabidopsis *chm*, *cue1*, and *im* mutants, and *Paphiopedilum concolor* [13,14,27,28,29,30]. In the comparison group VA0 vs. VA1, the *LHCB* subfamily genes and *RBCS* family genes did not have common DEGs in the green and variegated areas, suggesting that the downregulated expression of the common DEGs affected the whole leaflet of the variegated plants.

The common DEGs involved in photorespiration showed upregulated expression in both GY vs. VY and GA1 vs. VA1 comparison groups, including aldolase-type TIM barrel family genes (*GOX*) and peroxisomal photorespiratory enzyme family genes (*AGT*). The *AGT* (Tp57577_TGAC_v2_gene19354) was also upregulated in the VA0 vs. VA1 comparison group. Under the same growth conditions, photorespiration might be stronger in the variegated plants than in the green plants.

##### Redox and Nitrogen Metabolism Pathways

Four common DEGs were found in the redox pathway, including 2-oxoglutarate and Fe(II)-dependent oxygenase superfamily genes (*2OG*) and glutamate-cysteine ligase superfamily genes (*GSH*) (Figure 4; Appendix A). All of the four common DEGs above were upregulated in both the young and adult leaflets of variegated plants, indicating that the oxidative stress pathway of variegated plants of *T. pratense* was more active. There is only one common DEG related to nitrogen metabolism that encodes glutamate-ammonia ligases (*NodGS*), the expression of which was upregulated in both the GY vs. VY and GA1 vs. VA1 comparison groups (Figure 4; Appendix A). The upregulated expression of this gene suggested that enhanced protein catabolism and nitrogen transport were the reasons for the air space-type variegation formation.

##### Chlorophyll Metabolic Pathway

In leaf variegation classification, both air space-type and chlorophyll type were related to the chloroplast development pathway [1]. To determine whether the air space-type variegation color of *T. pratense* was related to chlorophyll deficiency, differential expression analysis of the tetrapyrrole pathway genes in chlorophyll synthesis was included in this study. As shown in Appendix A, among all of the DEGs, nine genes in the three comparison groups were annotated to the tetrapyrrole pathway, and no DEGs were common in the young and adult leaflets.

During the chlorophyll synthesis in the *T. pratense* leaflets, the DEGs were involved in the rate-limiting step of 5-aminolevulinic acid (ALA) synthesis and the chlorophyll branch Mg-protoporphyrin IX (Mg-PPIX) synthesis. Five of the DEGs in the GY vs. VY comparison group were annotated to the chlorophyll branch, encoding the fatty acid amide hydrolase family genes (4) and the Mg-chelatase H subunit (1), of which the downregulated expression of the largest H subunit (*GUN5*) of Mg-chelatase could block chlorophyll synthesis, indicating that chlorophyll synthesis was inhibited in the young leaflets of the variegated plants. In the GA1 vs. VA1 and VA0 vs. VA1 comparison groups, one DEG each was annotated to the glutamyl-tRNA reductase family (*HEMA2*), which belongs to the synthesis step of ALA synthesis. The protease encoded by *HEMA2* is induced in photosynthetic tissues under oxidative conditions, to supply heme for the defensive hemoproteins outside the plastids [31]. The differential expression of *HEMA2* in the adult leaflets of the variegated plants might be related to the response of the leaves to oxidative stress, while their downregulated expression might affect the heme synthetics of the *T. pratense* leaflets. The other two DEGs were annotated to the siroheme branch, which is not directly related to chlorophyll synthesis, indicating that chlorophyll synthesis in the variegated and green areas did not differ much at the transcriptional level. However, whether there are differences in chlorophyll content requires further study.

### 2.3. Leaf Variegation Associated Protein–Protein Network Interaction Analysis in the Young Leaflet Comparison Group

The leaf variegated mutants show differences in gene expression at the early stage of leaf development [13,26]. In this study, a protein–protein network interaction analysis was performed on the DEGs in the young leaflet comparison group, GY vs. VY. According to the corresponding Arabidopsis orthologs, the online STRING database was used to construct the protein–protein interaction network of 112 DEGs (Appendix A). These DEGs were annotated to the four pathways of cell wall development, photosynthesis, redox, and nitrogen metabolism (minimum interaction score was medium confidence score >0.4, MCL clustering).

The translated proteins of 60 DEGs were observed to interact, and three interacting protein clusters could be clearly distinguished (Figure 5). The first protein cluster was composed of the DEGs related to light reaction and the Calvin cycle. The light reaction part was composed of the LHCB subfamily proteins. The Calvin cycle part was composed of CP12-1, AT1G73110, AT3G01850, UKL5, RBCS1B, GAPA, FBP, FBA7, and AT3G60750 (Figure 5a). The LHCB subfamily proteins and the proteins translated from the genes related to the Calvin cycle were associated with multiple proteins involved in photorespiration, redox, and nitrogen metabolism in the second protein cluster, and there was also a complex interaction between photorespiration, redox, and nitrogen metabolism (Figure 5a,b). The third protein cluster was mainly composed of CSL, EXP, FLA, PG, PLL, PME, and XTH family proteins, which were involved in cell wall development, and involved multiple pathways, such as cell wall modification, pectin synthesis, and cell wall degradation (Figure 5c). Among the three interacting protein clusters, there was a complex interaction between the first protein cluster (light reaction and the Calvin cycle) and the second protein cluster (photorespiration, redox, and nitrogen metabolism). The Calvin cycle-related protein (FBP) in the first protein cluster interacted with the cell wall degradation-related protein (AT5G63180) and the cell wall modification-related protein (XTH31) in the third protein cluster. In addition, the Calvin cycle-associated protein (FBA7) in the first protein cluster interacted with the cell wall modification-associated protein (AT5G04310) in the third protein cluster. The cellulose synthesis-related protein (CSLB03) in the third protein cluster interacted with the Calvin cycle-related protein (AT1G73110) in the first protein cluster, as well as the photorespiration-related protein (GOX2) and the aerobic metabolism-related protein (AT2G41220) in the second protein cluster. The interactions of these protein clusters can demonstrate the association between the genes involved in the light reaction and the Calvin cycle and the genes involved in cell wall development.

### 2.4. Relationship between Protease Degradation Systems and Leaf Variegation Formation

All of the eukaryotes depend on the ubiquitin-proteasome system and autophagy to control the abundance of the key regulatory proteins and maintain a healthy intracellular environment [32]. To explore the relationship between protease degradation systems and leaf variegation formation, we analyzed ubiquitin- and autophagy-dependent degradation in the three comparison groups of *T. pratense*. A total of 122 DEGs were annotated to the ubiquitin protease pathway. There were 87, 56, and 16 DEGs in the three comparison groups, and no DEG was annotated to the autophagy pathway (Figure 6; Appendix A).

During air space-type variegation formation in *T. pratense*, no DEG was annotated to the ubiquitin-activating enzyme (E1) in the three comparison groups. The ubiquitin-conjugating enzyme (E2) had only two DEGs in the young leaflet comparison group GY vs. VY, and its expression, was upregulated (Figure 6a). The enzyme has conjugating- protein ligase activity. The ubiquitin-protein ligase enzyme (E3) directly removes the damaged proteins through the degradation pathway and participates in the protein quality-control pathway. At the same time, it also removes damaged proteins, chloroplasts, and other organelles in batches by regulating the key factors of autophagy and cell death [33,34,35], and participates in cell death regulation. The DEGs of the three comparison groups were annotated to the E3 ligase enzyme. There were 77 DEGs in the GY vs. VY comparison group and 47 DEGs in the GA1 vs. VA1 comparison group, and there were only 15 DEGs in the comparison group VA0 vs. VA1 (Figure 6; Appendix A). In the different comparison groups, the gene expression of the different types of subunits encoding the E3 ligase enzyme was also different. The DEGs of the GY vs. VY comparison group were annotated to DCX type, HECT type, RING type, and SCF type (FBOX and SKP) subunits. The number of the DEGs of RING type and SCF type subunits was the largest, with 75 DEGs showing mainly upregulated expression. The DEGs of the GA1 vs. VA1 comparison group were annotated to the APC type, DCX type, RING type, and SCF type subunits. Similarly, the number of DEGs of the RING type and SCF type subunits was the largest, with 45 DEGs showing mainly upregulated expression. In the VA0 vs. VA1 comparison group, DEGs were only annotated with the RING type and SCF type (FBOX) subunits, with 15 DEGs showing mainly downregulated expression. The number of chloroplasts in the *T. pratense* leaf cross-section was lower in the variegated plants than in the green plants, while the number of chloroplasts in the variegated area was not significantly different from that in the green area (Appendix A). Thus, the decrease in the chloroplast number in the variegated plants might be related to the modification and degradation of ubiquitin ligase. The E3 ligase enzyme in the variegated plants plays a stronger role in protein degradation, organelle modification and degradation, and cell death than in the green plants, which might contribute to the intercellular white variegation formation. However, no significant difference in the influence of chloroplasts was found between the variegated and green areas of the variegated plants. Therefore, the color change caused by the difference in the number of chloroplasts between the variegated and green areas had little to no effect on the color development of the air space-type variegation in *T. pratense*.

In the deubiquitinating enzymes (DUB), two DEGs were annotated to the encoding ubiquitin-specific protease. In the GA1 vs. VA1 and VA0 vs. VA1 comparison groups, the expression of only one common DEG was downregulated, but no DEG was found in the comparison group VA0 vs. VA1. Thus, the deubiquitination of the variegated plants was inhibited in the young and adult leaflets, which increased the ubiquitination of the abnormal proteins, and then regulated the abnormal changes in the air space-type variegation in *T. pratense*.

## 3. Discussion

### 3.1. Photosystem-Related Genes Play an Important Role in Air Space-Type Variegation Formation in T. pratense

In the bioinformatics analysis of the *T. pratense* leaflets, we found that the downregulated *LHCB* subfamily gene was one of the main reasons for the air space-type variegation formation. In plants, the LHCB proteins encoded by the *LHCB* subfamily genes are abundant membrane proteins in photosystem II and are involved in light energy capture, the regulation and distribution of excitation energy, maintenance of thylakoid membrane structure, photoprotection, and the response to gene changes during cell wall development [36,37,38,39,40,41,42,43]. Among the Arabidopsis air space-type variegation mutants, the *LHCB* subfamily gene expression of the *chm* mutants [30,44], *cla1* mutants [45], *cue1* mutants [29,46] and *im* mutants [13] was downregulated during leaf variegation formation, and chloroplast deletion was found in the palisade cells of the variegated areas. At the same time, the *LHCB* subfamily genes were also downregulated in the herbicide-induced air space-type variegation in wild-type Arabidopsis, and chloroplasts were absent in the palisade cells of the variegated areas [13]. In the transcriptome study of the naturally formed air space-type variegation in *Paphiopedilum concolor*, the *LHCB* subfamily genes (*light-harvesting complex* and *chlorophyll a-b binding protein*, *CP24 10A*) involved in the light response also showed downregulated expression [14].

In this study, the *LHCB* subfamily genes showed downregulated expression in the young and adult leaflets of the variegated plants in *T. pratense* (Figure 4). In the cross-section of the leaflets, the number of chloroplasts was significantly lower in the variegated plants than in the green plants (Appendix A). Unlike the leaf variegations mutant of Arabidopsis, the number of chloroplasts in the variegated areas and green areas of the variegated plants in *T. pratense* was not significantly different. The *LHCB* subfamily genes also did not show a downregulated expression in the comparison group of green areas and variegated areas (VA0 vs. VA1) of the adult leaflets of the variegated plants (Figure 4 and Appendix A). Therefore, the downregulated *LHCB* subfamily gene expression might affect the cell development of the whole *T. pratense* leaflet. Abnormal chloroplast development affects the intercellular trafficking by plasmodesmata [47] and reduces palisade cell elongation [48,49]. In the signal transduction pathway of chloroplast genesis, the *LHCB* subfamily genes are a key component and play important roles in plant photosynthesis and chloroplast grana formation [50,51], indicating that the downregulated *LHCB* subfamily gene expression affects the formation of normal chloroplast morphology. In conclusion, this study suggests that air space-type variegation formation in *T. pratense* was closely related to the downregulated expression of the *LHCB* subfamily genes.

However, some of the gene mutations can directly affect the palisade cell development without inhibiting *LHCB* subfamily gene expression and produce variegations with gaps, such as the Arabidopsis *pac* mutants [52,53] and *var3* mutants [54]. Under a strong light environment, the white areas easily spread to the whole leaf, and few or missing palisade cells in the variegated areas resulted in gaps and a rough paraxial surface in the variegated area [52,53,54]. In the studies of macro morphology and cross-sectional views of air space-type variegated leaf plants, the macroscopic leaf variegation patterns are stable, and the paraxial surface of the variegated area does not become rough and wrinkled due to morphological changes in the palisade cells [1,55]. Obviously, the mechanism of this kind of leaf variegation formation in Arabidopsis *pac* and *var3* mutants differed from the natural mechanism of air space-type variegation formation. Leaf cross-section observations have shown that chloroplast deletion also exists in the variegated areas of air space-type variegated leaf plants, such as *Blastus cochinchinensis* [55] and *Sonerila cantonensis* [1]. Therefore, inhibiting the *LHCB* subfamily gene expression to affect air space-type variegation formation may widely exist in nature.

### 3.2. Upregulated Expression of Genes Involved in Cell Wall Structure Modification Is Important for Air Space-Type Variegation Formation in T. pratense

The upregulated expression of genes involved in cell wall structural modification can affect the palisade cells’ normal formation in *T. pratense* variegated plants. In the cross-section of the leaflets of *T. pratense*, significant differences were found in the palisade cell morphology in the variegated area compared with the green area and the corresponding area in the green plants (Appendix A). The results of the KEGG enrichment analysis of different comparison groups of *T. pratense* showed that the DEGs of the three comparison groups were significantly enriched in flavonoid biosynthesis and isoflavone biosynthesis pathways (Figure 2). These pathways are involved in cell wall synthesis [21,22,23,24,25]. In the subsequent MapMan analysis, the number of DEGs involved in cell wall development was also the largest among the four pathways (Figure 3 and Figure 4; Appendix A), which highlights the importance of the DEGs related to cell wall development in air space-type variegation. The cell wall can affect the size and shape of cells, as well as control the cell and tissue structure [56,57,58,59], indicating that the air space-type variegation formation in *T. pratense* was closely related to the development of the cell wall structure. The cell wall structure analysis of the Arabidopsis *im* mutant leaves showed that the cell walls in the white areas have reduced lignin and cellulose microfibrils, as well as changes in galactomannans and the decoration of xyloglucan, which differed from the green areas and the wild type [26]. The air space-type variegation formation was also related to the change in the cell wall structure.

During plant growth and development, the cell wall enzymes play important roles in cell wall remodeling and metabolism [60,61]. Eleven screened gene families (*CESA*, *CSL*, *EXP*, *FLA*, *PG*, *PGIP*, *PLL*, *PME*, *RGP*, *SKS*, and *XTH*) related to air space-type variegation formation were involved in the structural modification of the cell wall and showed upregulated expression in the variegated plants (Figure 4; Appendix A). The upregulated expression of the genes from these families often causes cell wall loosening and strength reduction [56,62,63,64,65,66], providing the appropriate conditions for the production of the intercellular spaces in *T. pratense* palisade tissue.

Eleven gene families related to cell wall structure modification were involved in different cell wall structure modification processes. The *CESA*, *CSL*, *FLA*, *PGIP*, *PG*, *PLL*, and *SKS* family genes can affect the integrity and structural characteristics of the cell wall and change the adhesion state of cells [67,68,69,70,71,72,73,74,75]. Expansin encoded by *EXP* is a key regulator of cell wall extension during plant growth that upregulates expression and can promote cell wall relaxation and extension [76,77,78]. *PME* encodes the plant invertase/pectin methylesterase inhibitor that catalyzes pectin production, which not only hardens the cell wall but also enhances the activities of polygalacturonase and pectin lyase to promote the degradation of pectin and cause cell wall relaxation [79,80]. *RGP* encodes a reversibly glycosylated polypeptide, which is an intercellular fibronectin. It participates in the regulation of polysaccharide biosynthesis in the cell wall and is involved in intercellular material transport [81,82,83,84,85,86]. Xyloglucan endotransglucosylase/hydrolase encoded by *XTH* can rapidly loosen the cell wall by breaking the load-bearing xyloglucan chains. An overexpression in Arabidopsis leaves produces intercellular spaces between the palisade tissue cells [87,88,89]. The leaflets of the variegated plants in *T. pratense* did not show variegation at the young leaflet stage, but variegation was obvious at the adult leaflet stage (Appendix A). In both the GA1 vs. VA1 and VA0 vs. VA1 comparison groups, the upregulated expression of *EXP* and *XTH* family genes could generate intercellular spaces [76,77,78,87,88,89]. The upregulated gene expression of these two families might be the reason for the changes in the cell wall structure, stress capacity, and adhesion state of *T. pratense*.

Hence, the formation of conspicuous intercellular spaces between the epidermal and palisade cells and among the palisade cells in the variegated area of *T. pratense* was affected by the upregulated expression of genes related to cell wall structural modification. Among the many genes related to cell wall structural modification, the role of *EXP* and *XTH* family genes in leaf variegation formation in *T. pratense* deserves special attention.

### 3.3. Potential Regulatory Network of Air Space-Type Variegation in T. pratense

Aluru et al. analyzed the changes in the main transcripts in the variegated area of the Arabidopsis *im* mutant and found that the expression of the photosynthesis-related genes was downregulated by *IM* gene mutation, and the expression of the mitochondria and peroxisome-related genes in the variegated area was upregulated [13]. This regulation mode was closely related to the influence of photooxidation on leaves during airspace-type variegation formation. Referring to the transcript changes in the *im* mutant and the transcript analysis results related to air space-type variegation formation in *T. pratense* (Figure 4; Appendix A), the possible regulation process in the V-shaped leaf variegation formation in *T. pratense* was as follows.

At the stage of unexpanded young leaflets of variegated *T. pratense* (GY vs. VY), the leaflets wrapped by membranous stipules were in a low-light environment and leaf variegation had not yet appeared, which was the same as the phenotype in the green plants (Appendix A). The Arabidopsis *im* mutant had the same phenotype in a low-light environment [90]. At this stage, the expression of the genes involved in the light reaction (*LHCB* subfamily genes) and the Calvin cycle (*RBCS* family genes) in variegated plants was inhibited, indicating that the chloroplast development in the young leaflets of the variegated plants might be affected. The genes related to photorespiration, redox, and nitrogen metabolism (*GOX*, *AGT*, *2OG*, *GSH*, and *NodGS* family genes) showed upregulated expression, suggesting that photorespiration in the young leaflets might be stronger in the variegated plants than in the green plants. At this time, the cell wall structure modification genes also showed upregulated expression (Figure 4). In response to oxidative stress, the early cell walls of the leaves can reshape leaf development by the rearrangement of its entire structure, to maintain normal chloroplast development [26,91,92]. The analysis of the leaf variegation-associated protein interaction network in the young leaflets showed interaction protein clusters between photosynthesis and the cell wall (Figure 5). The upregulated expression of the cell wall structure modification genes at the young leaflet stage of the variegated plants was related to maintaining normal chloroplast development, suggesting the presence of abnormal chloroplasts in mesophyll cells at the young leaflet stage of the variegated plants.

When the leaflets of the variegated plants were fully expanded (adult leaflet stage), they were exposed to direct sunlight, producing the V-shaped pattern. At this stage (GA1 vs. VA1), the genes related to light reaction and the Calvin cycle in variegated plants showed a downregulated expression, and the genes related to photorespiration, redox, nitrogen metabolism, and cell wall structure modification showed an upregulated expression (Figure 4). The KEGG enrichment analysis showed that the two comparison groups (GA1 vs. VA1 and VA0 vs. VA1) at the adult leaflet stage of the variegated plants had significantly enriched DEGs in response to stress, reactive oxygen species-related metabolic pathways, and photorespiration in photosynthesis (Figure 2), indicating that the adult leaflets of the variegated plants might have active secondary metabolism and stress responses, suggesting that the leaflets were subjected to oxidative stress. The leaf cross-section observations also showed that the chloroplasts in adult leaflets were fewer in the variegated plants than in the green plants (Appendix A). At the same time, the coding genes of E3 ubiquitin ligase-related subunits involved in protein degradation, autophagy, and cell death were significantly upregulated in the GY vs. VY and GA1 vs. VA1 comparison groups (Figure 6). Based on the above results, a series of organelle clearance and cell death events caused by oxidative stress might occur in the leaflets of the variegated *T. pratense* under strong light. In the Arabidopsis leaf variegation mutation with photooxidation, the chloroplast development defect is more serious with the increase in light intensity [93,94]. Therefore, this study suggests that air space-type variegation formation in *T. pratense* conforms to the photooxidation hypothesis. *LHCB* subfamily gene expression was downregulated, and the number of chloroplasts was lower in the cells of the variegated plants (variegated areas and green areas) than in the cells of the green plants, which reflected the decrease in photosynthetic activity. The relationship between the enhancement of the stress response and the decrease in photosynthetic activity in the variegated plants was difficult to determine, but they were closely related to leaf variegation formation.

Leaf development involves two main stages. The first stage was dominated by cell proliferation and the second stage was dominated by cell expansion [95,96]. From the never unfolded young leaflet stage to the fully expanded adult leaflet stage of *T. pratense*, the cells changed from division to expansion. Because of the upregulated expression of genes related to the cell wall structural modification, the strength and stress of the cell wall were changed. Under the influence of cell expansion, the palisade cell morphology in the whole leaflet (variegated area VA1 and green area VA0) was changed to form non-cylindrical palisade cells. The palisade cell morphology differed from the corresponding area (GA1) in green plants (Appendix A). The DEGs related to photosynthesis, redox, nitrogen metabolism, and cell wall development were also the least in the VA0 vs. VA1 comparison group, and hardly any of the DEGs were related to photosynthesis, redox, and nitrogen metabolism (Figure 4; Appendix A), indicating that the differential expression of these genes could impact the whole leaflet of variegated plants. In the protease degradation system, the DEGs of the VA0 vs. VA1 comparison group were also the least (Figure 6c), indicating that the regulation of the protease degradation system in both green and variegated areas was not significant at the adult leaflet stage of the variegated plants. These evidences support that the effect of leaf variegation development on variegated plant leaflets is holistic. Compared with the green area, the genes related to cell wall structure modification in the variegated area were significantly upregulated, indicating that the changes in cell wall strength and stress were more obvious in the variegated area than in the green area (Figure 4; Appendix A). Under the influence of the genes related to cell wall structural modification and intermediary growth ability, the change in the palisade cell morphology in the variegated area was stronger than that in the green area, resulting in the increase in intercellular spaces. The intercellular spaces might cause total reflection when light passes through the speckled area on the paraxial surface of the blade. In the macro form, the blade presents white speckles (Appendix A). In the three comparison groups, nine DEGs were related to the tetrapyrrole pathway (Appendix A), and the expression of only one key gene of chlorophyll synthesis was downregulated at the young leaflet stage of the *T. pratense* variegated plants, indicating that chlorophyll synthesis at the young leaflet stage of variegated plants was affected, but the young leaflets of the *T. pratense* variegated plants did not show V-shaped variegation. No key genes for chlorophyll synthesis were found in the adult leaflets. Therefore, the change in chlorophyll content might affect the overall leaf color of variegated plants and had no effect on the color development of air space-type variegation in *T. pratense*.

Leaf expansion is mainly driven by turgor pressure that spreads along the longitudinal and transverse axis. The main vein affects the longitudinal and transverse expansion. At a local level, the difference in stress or different characteristics of the cell wall will theoretically lead to cell growth at different speeds [97,98,99]. The biological and physical balance between the internal expansion pressure of cells and the stress on the cell wall is the key factor for plant cell growth [100]. Even with enough metabolic energy and raw materials to produce new cytoplasm, cell membranes, and cell walls, as well as sufficient turgor pressure to drive expansion, the only factor inhibiting cell growth is the cell wall [101]. Under the influence of the main vein, the variegated *T. pratense* plants leaflets extended faster at the apex along the direction of the main vein rather than the lateral vein, increasing the stress on the cells at the center of the leaflets. At the same time, the expression of genes related to the structural modification of the cell wall in the variegated area of the leaflets was upregulated, which caused the loosening and hardness of the palisade tissue cell walls, and changed the cell growth rate. Hence, under the influence of the expansion pressure and the change in the force on the cell wall, the shape of the palisade cells, that originally grew evenly, changed, forming a V-shaped pattern on the leaf paraxial surface with the V opening facing toward the leaflet base (Figure 7).

Among the air space-type variegated leaf plants, some species had the same shape as the leaf variegation of *T. pratense*, showing the V-shape in the center of the leaf. For example, the cross-sectional structure of the leaves of *Persicaria chinensis* and *Amaranthus viridis* was similar to that of *T. pratense*, the shape of the palisade cells in the variegated area changed, and the chloroplast content in the variegated area was not significantly different from that in the green area [1]. A white variegation of *P. chinensis* that was obvious in normal light did not appear in the low-light environment. Judging from the commonalities with leaf variegation in *T. pratense* in terms of morphology and light adaptation, the regulation mode of air space-type variegation in *T. pratense* might also be applicable to these species.

## 4. Materials and Methods

### 4.1. Plant Materials and Growth Conditions

The *T. pratense* seeds were purchased from Shenzhen Huacaoyuan Seed Industry (accessed on 1 June 2017, http://zzd.mmzzx.com/) and were planted in the field outside the greenhouse of the Biological Station of East China Normal University. The variegated and green plants with stable characters were selected from germinating individuals as research material. Firstly, these plants were moved to separate greenhouses with the same maintenance conditions to avoid pollen exchange between the two phenotypes. Secondly, the seeds of the two phenotypes were collected after fruiting and were planted in hole trays. The sowing substrate was a 1:1 mixture of nutrient soil and vermiculite. Finally, the seedlings of well-grown variegated and green plants were selected after germination and transplanted into large pots. The cultivation substrate was the same as the sowing substrate, and the plants were placed in the same greenhouse under natural light. After 5 months of growth, the plant height was about 30 cm, and the transcriptome sequencing materials were sampled when plants had four–five branches.

### 4.2. RNA-Seq Material Collection

Three variegated and three green similar-sized plants of *T. pratense* grown under the same conditions were selected. Five parts (Appendix A) of young leaflets (unexpanded leaflets) and adult leaflets (fully expanded leaflets) of the variegated and green plants were collected (green plant young leaflets GY, variegated plant young leaflets VY, green area of variegated plant adult leaflets VA0, white area of variegated plant adult leaflets VA1, and the position corresponding to VA1 of green plant adult leaflets GA1).

A double-sided blade was used to cut the material. At the young leaflet stage, only petioles and petiolules were removed and the intact leaflets were retained. When taking materials from the adult leaflets, only the clear white parts were retained in the variegated area (VA1) of variegated plants, and the parts with a blurred speckled boundary were removed. Parts at the junction of the variegated area and leaf margin were removed from the green area (VA0). For green plants, the position corresponding to VA1 on the green plant adult leaflets (GA1) was collected. The specific sample parts are shown in Appendix A. Five parts (GA1, GY, VA0, VA1, and VY, three replicates) were sampled. Each sample was mixed with materials from the same area of 20 leaflets of the same plant. After the material was collected, it was immediately frozen in liquid nitrogen, wrapped with aluminum foil, and stored in the freezer at −80 °C. There were three comparison groups: (1) GY was the control group, and VY was the experimental group (GY vs. VY); (2) GA1 was the control group, and VA1 was the experimental group (GA1 vs. VA1); and (3) VA0 was the control group, and VA1 was the experimental group (VA0 vs. VA1).

### 4.3. Transcriptome Sequencing and Preliminary Data Processing

#### 4.3.1. mRNA Extraction, Quality Detection, and cDNA Library Construction

Fifteen samples were sent to BGI (Shenzhen, China) for mRNA extraction, sequencing, and library construction. The procedure was as follows: tissues of 15 samples from five parts of *T. pratense*, viz. GA1, GY, VA0, VA1, and VY, were pulverized, and the total RNA of each sample was extracted using the TRIzol method. Next, the mRNA was enriched by oligo-dT magnetic beads, and eluted. Finally, purified *T. pratense* mRNA was obtained using the purification kit.

The mRNA concentration, OD260/280, OD260/230, and RNA integrity number (RIN) were detected by the Agilent 2100 bioanalyzer (Agilent Technologies, Santa Clara, CA, USA) and the NanoDrop spectrophotometer (Thermo Fisher Scientific, Waltham, MA, USA). The samples with RIN ≥ 7.0 were used for library construction.

#### 4.3.2. RNA-seq, Assembly, and Functional Annotation

The single-stranded circular DNA library was sequenced using the DNBSEQ platform to obtain an RNA-seq dataset. The raw reads were filtered with SOAPnuke software [102] to remove unqualified data (low-quality reads, sequencing adapters, and sequences containing more than 5% unknown bases) to obtain high-quality *T. pratense* clean reads.

The clean reads were compared with the *T. pratense* genome [17] using HISAT2 software [103], and the transcripts were reconstructed with StringTie software [104]. CPC software [105] was used to predict the protein-coding potential of the new transcripts, and Bowtie 2 software [106] was used to compare the new transcripts to the known gene/protein sequence databases of GO, KEGG, NR/NT, Swiss-Prot, Pfam, KOG, PlantTFDB, and PRGdb (Appendix A). The expression of the whole transcript was determined using RSEM software [107].

### 4.4. Screening and Identification of DEGs Related to Air Space-Type Variegation Formation

The DEGs of the three comparison groups (GY vs. VY, GA1 vs. VA1, and VA0 vs. VA1) of *T. pratense* were evaluated with DEGseq software [108]. The KEGG enrichment analysis was performed on DEGs using the phyper function in R. The DEGs of the three comparison groups were annotated with the Mercator software [109], followed by metabolic pathway analysis performed using MapMan software [110]. Finally, the Venn analysis online tool (accessed on 6 April 2022, https://bioinfogp.cnb.csic.es/tools/venny/) was used to screen the DEGs related to air space-type variegation formation in *T. pratense*. The heatmaps were drawn using the HeatMap plug-in on the TBtools platform [111]. The protein interaction analysis was performed using the online STRING database (accessed on 13 January 2022, https://cn.string-db.org/cgi/input.pl). The software versions and parameters are shown in Appendix A).

## Figures and Tables

**Figure 1 ijms-23-07794-f001:**
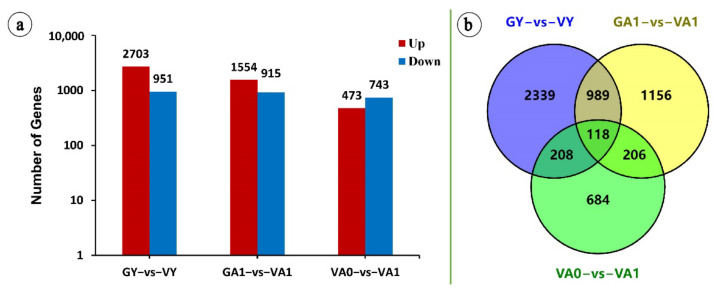
Statistics of DEGs (**a**) and Venn analysis (**b**) of the three comparison groups. Red, upregulated; blue, downregulated. Y-coordinate value, log base 10.

**Figure 2 ijms-23-07794-f002:**
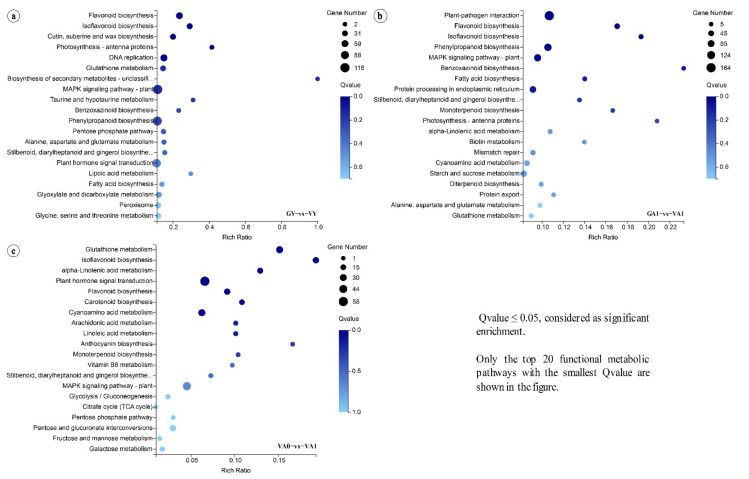
Enrichment of DEGs in the KEGG pathways for the three comparison groups. (**a**) for GY vs. VY; (**b**) for GA1 vs. VA1; (**c**) for VA0 vs. VA1.

**Figure 3 ijms-23-07794-f003:**
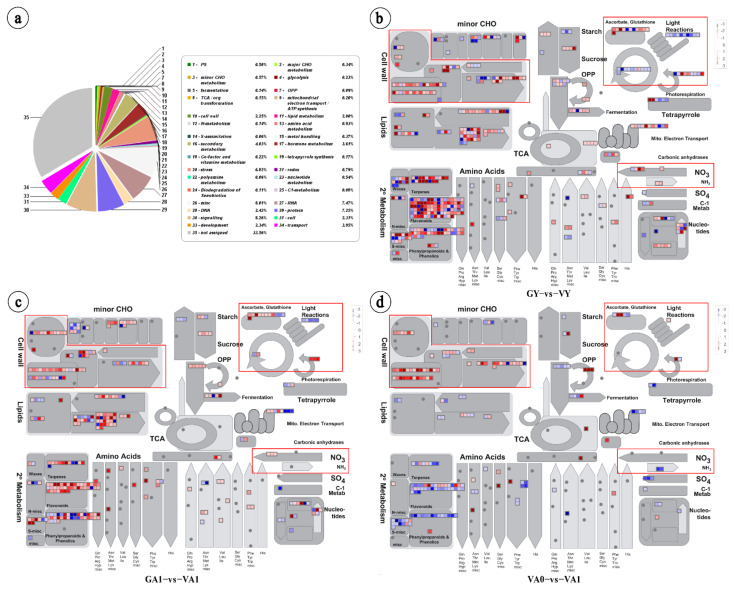
Mercator annotation results and metabolism overview for three comparison groups of *Trifolium pratense*. (**a**) Mercator annotation results; (**b**) visualization of metabolism overview in GY vs. VY; (**c**) visualization of metabolism overview in GA1 vs. VA1; (**d**) visualization of metabolism overview in VA0 vs. VA1; red, upregulated; blue, downregulated.

**Figure 4 ijms-23-07794-f004:**
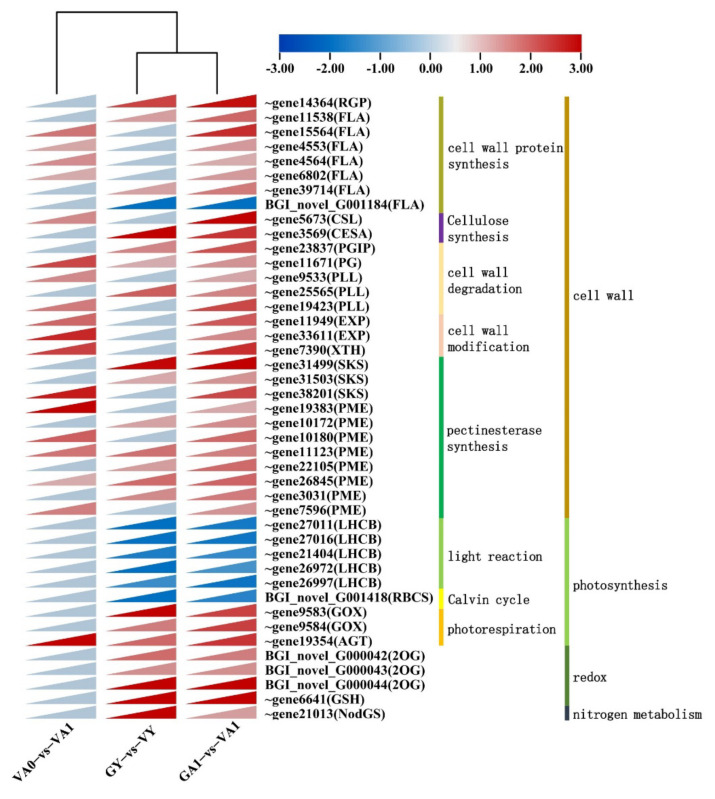
Heatmaps of DEGs in the three comparison groups. Red, upregulated; blue, downregulated. ~, Tp57577_TGAC_v2_.

**Figure 5 ijms-23-07794-f005:**
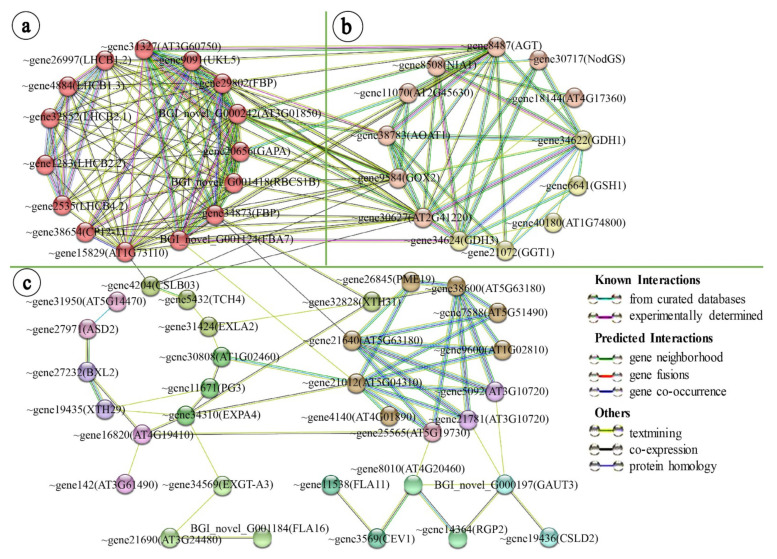
Protein–protein interaction network of DEGs in the young leaflet. (**a**) light reaction and Calvin cycle; (**b**) photorespiration, redox, and nitrogen metabolism; (**c**) cell wall development. Line colors indicate the type of interactive evidence; ~, Tp57577_TGAC_v2_.

**Figure 6 ijms-23-07794-f006:**
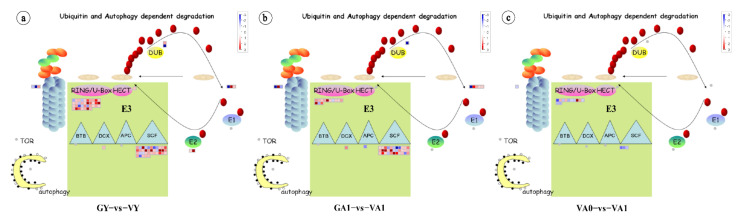
Visualization of ubiquitin and autophagy dependent degradation of DEGs in the three comparison groups. (**a**) Visualization of DEGs in the GY vs. VY; **(b**) Visualization of DEGs in the GA1 vs. VA1; (**c**) Visualization of DEGs in the VA0 vs. VA1. Red, upregulated; blue, downregulated. The graph was generated by MapMan software.

**Figure 7 ijms-23-07794-f007:**
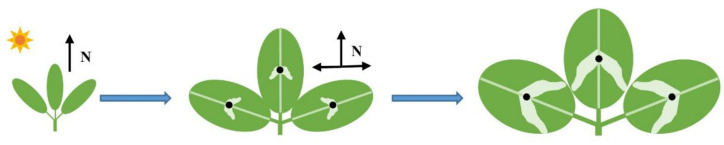
Development of V-shaped leaf variegation in *Trifolium pratense*. N, direction of force.

**Table 1 ijms-23-07794-t001:** RNA-seq and alignment of *Trifolium pratense* variegated leaf samples.

Sample	Raw Reads (M)	Clean Reads (M)	Clean Bases (Gb)	Q20	Q30	Mapped Genome	Mapped Genes
GY_1	47.43	43.17	6.48	96.69%	88.28%	78.99%	69.76%
GY_2	47.43	42.90	6.43	96.58%	88.02%	79.16%	70.29%
GY_3	47.43	43.22	6.48	96.74%	88.37%	79.70%	71.19%
VY_1	49.19	44.12	6.62	96.69%	88.35%	80.04%	67.25%
VY_2	47.43	42.55	6.38	96.84%	88.70%	79.78%	67.26%
VY_3	49.19	44.66	6.70	96.77%	88.53%	80.14%	69.17%
GA1_1	47.43	43.62	6.54	96.67%	88.18%	77.55%	70.51%
GA1_2	47.43	43.97	6.60	96.78%	88.47%	77.79%	71.49%
GA1_3	49.19	44.40	6.66	96.81%	88.59%	78.43%	70.59%
VA0_1	51.96	47.11	7.07	96.55%	88.44%	77.54%	69.49%
VA0_2	52.70	47.92	7.19	96.69%	88.75%	78.37%	68.71%
VA0_3	52.70	47.72	7.16	96.70%	88.66%	79.03%	69.88%
VA1_1	52.47	47.49	7.12	96.78%	88.94%	78.22%	69.17%
VA1_2	51.19	46.95	7.04	97.50%	90.33%	79.91%	70.93%
VA1_3	47.39	43.02	6.45	95.85%	87.14%	76.59%	69.41%

Note: M, one million.

## Data Availability

The data that support the findings of this study have been deposited in the CNSA (accessed on 28 June 2022, https://db.cngb.org/cnsa/) of CNGBdb with accession code CNP0003184.

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
