# Peer review of "Transcriptome Analysis of Air Space-Type Variegation Formation in Trifolium pratense"

_ijms, 2022, doi:10.3390/ijms23147794_

Round 1
Reviewer 1 Report
In this study, Zhang et al. examined V-shaped air space type variegation in Trifolium pratense. The study is well performed and the description is easy to follow. The Introduction contains the necessary background information, the presented results will be of interest to specialists in the field of plant physiology. The Discussion section provides a comprehensive interpretation of the results in relation to the relevant literature. I believe that the manuscript can be published after correcting technical flaws:
Numerical references to cited articles should be placed in brackets […] as indicated in the rules for the preparation of the manuscript.
Lines 52, 413, 526: Notices should be removed and correct references should be provided.
Line 56: Reference "18-1920" should be corrected to "[18,19,20]".
Line 163: The sentence is not finished.
Lines 178, 192, 242, 336, 358, 368, 403, 408, 410, 416, 423, 454, 488, 524: References are given in an incorrect form. Correct reference numbers should be given with appropriate punctuation.
The quality of figures 2, 3, 5, 6 should be improved.
Author Response
Dear Reviewer,
Thank you for comments on our manuscript entitled “Transcriptome Analysis of Air Space Type Variegation Formation in Trifolium pratense”. We resubmitted the manuscript with detailed interpretation to the comments and revised the manuscript carefully for some vital problems put forward.
The vital problems include 1) corrected citation of references; 2) replaced with higher quality pictures; 3) revised the whole manuscript carefully and tried to avoid any grammar or syntax error.
Hope it meet the criterion of International Journal of Molecular Sciences.
Sincerely,
Contact person:
Jianghang Zhang
Shanghai, China
School of Life Sciences, East China Normal University, Shanghai 200241, China
E-mail: 783648156@qq.com
Corresponding author:
Hongqing Li
Shanghai, China
School of Life Sciences, East China Normal University, Shanghai 200241, China
E-mail: hqli@bio.ecnu.edu.cn
Reply to reviewers' comments:
1) Numerical references to cited articles should be placed in brackets […] as indicated in the rules for the preparation of the manuscript.
Answer: Done. We have corrected citation of references.
2) Lines 52, 413, 526: Notices should be removed and correct references should be provided.
Answer: Done.
3) Line 56: Reference "18-1920" should be corrected to "[18,19,20]".
Answer: Done.
4) Line 163: The sentence is not finished.
Answer: Yes. We revised the words
5) Lines 178, 192, 242, 336, 358, 368, 403, 408, 410, 416, 423, 454, 488, 524: References are given in an incorrect form. Correct reference numbers should be given with appropriate punctuation.
Answer: Done. We have revised the whole manuscript carefully.
6) The quality of figures 2, 3, 5, 6 should be improved.
Answer: We have replaced with higher quality pictures.
Reviewer 2 Report
Review of ijms 1751573:
The manuscript by Zhang et al., Transcriptome Analysis of Air Space Type Variegation Formation in Trifolium pratense is nice, comprehensive transcriptome work on the differential gene expression of the red clover chevron variegation from young and older leaves. While the manuscript was difficult to understand to this reviewer as it was not composed by UK or US English speaking person, the correct wordage is used. The molecular and bioinformatic presentation followed essentially what was provided from the BGI bioinformatic pipeline. However, the authors chose pertinent and not-surprising metabolic pathways (cell wall development, photosynthesis, redox and nitrogen metabolism) to highlight, and integrated the results with previous findings from the literature nicely. The authors did a good job integrating the relevant background work and included (probably more than needed) the pertinent references. While this reviewer is not familiar with the ecological relevance of the studied variegation in nature, it was an interesting read, and the authors did a good job in the discussion including the information. Aside from possible editorial issues outlined above, this reviewer has no specific issues with the presentation of the manuscript except for some editorial problems associated with the PDF document sent out for review by IJMS detailed below. One final comment: Have the authors submitted, or plan to submit, the raw read files to a public repository? I did not find reference to such.
Minor comments:
Ln555-556: as parents or as selected individuals for sampling? Parents implies that these plants were used for crossing and I did not see any reference to genetic analysis.
Ln580: This is a little confusing. Maybe simply state that three individual plants were sampled and in parenthesis (treated as biological replicates)…
Ln582: a -80 is not a refrigerator, it is a freezer.
Comments to IJMS editorial board:
References were not correctly referenced within the manuscript. While it was not difficult to discern, this needs to be corrected. The figures presented in the pdf-for-review (i.e. Figure 2, 3, 5, 6 ) probably derived from the BGI .jpg files, had lettering that was too small (gritty pixels) to discern the fine print. The journal needs to send out “corrected” pdf’s for review, however, this appears to be an endemic issue with MDPI journals in general. It would be helpful to provide .html files, and for figures regular sized .jpg, .tiff, the BGI .png files.
Author Response
Dear Reviewer,
Thank you for comments on our manuscript entitled “Transcriptome Analysis of Air Space Type Variegation Formation in Trifolium pratense”. We resubmitted the manuscript with detailed interpretation to the comments and revised the manuscript carefully for some vital problems put forward.
The vital problems include 1) corrected citation of references; 2) replaced with higher quality pictures; 3) revised the whole manuscript carefully and tried to avoid any grammar or syntax error.
Hope it meet the criterion of International Journal of Molecular Sciences.
Sincerely,
Contact person:
Jianghang Zhang
Shanghai, China
School of Life Sciences, East China Normal University, Shanghai 200241, China
E-mail: 783648156@qq.com
Corresponding author:
Hongqing Li
Shanghai, China
School of Life Sciences, East China Normal University, Shanghai 200241, China
E-mail: hqli@bio.ecnu.edu.cn
Reply to reviewers' comments:
1) Have the authors submitted, or plan to submit, the raw read files to a public repository? I did not find reference to such.
Answer: The data that support the findings of this study have been deposited in the CNSA (https://db.cngb.org/cnsa/) of CNGBdb with accession code CNP0003184. But it may take some time to retrieve.
2) Ln555-556: as parents or as selected individuals for sampling? Parents implies that these plants were used for crossing and I did not see any reference to genetic analysis.
Answer: Thanks. As selected individuals for sampling. The wording here is incorrect and has been corrected.
3) Ln580: This is a little confusing. Maybe simply state that three individual plants were sampled and in parenthesis (treated as biological replicates)…
Answer: Thanks. We revised the words.
4) Ln582: a -80 is not a refrigerator, it is a freezer.
Answer: Done. Thanks.
5) The figures presented in the pdf-for-review (i.e. Figure 2, 3, 5, 6) probably derived from the BGI .jpg files, had lettering that was too small (gritty pixels) to discern the fine print.
Answer: Done. We have replaced with higher quality pictures.
6) Moderate English changes required
Answer: Done. We have revised the whole manuscript carefully and tried to avoid any grammar or syntax error.